# Treatment Strategies for Locoregional Recurrence in Esophageal Squamous-Cell Carcinoma: An Updated Review

**DOI:** 10.3390/cancers16142539

**Published:** 2024-07-14

**Authors:** Atsushi Mitamura, Shingo Tsujinaka, Toru Nakano, Kentaro Sawada, Chikashi Shibata

**Affiliations:** Division of Gastroenterological Surgery, Department of Surgery, Tohoku Medical and Pharmaceutical University, Sendai 983-8536, Japan

**Keywords:** chemotherapy, chemoradiotherapy, esophageal squamous-cell carcinoma, immune checkpoint inhibitors, radiotherapy, salvage surgery, triplet regimen

## Abstract

**Simple Summary:**

Despite recent advancements in treating advanced esophageal squamous-cell carcinoma, the outcomes remain uncertain due to the high rates of cancer returning locally and spreading to distant areas. While surgery can be effective when cancer recurs in the lymph nodes of the neck, its success is limited if the cancer recurs in multiple areas. For cases that cannot be operated on or have recurred, radiotherapy and chemoradiotherapy are used, with recent technical improvements reducing the side effects. Proton beam therapy is also an option. Immune checkpoint inhibitors, such as pembrolizumab and nivolumab, alongside chemotherapy, have resulted in improved survival rates in these challenging cases. Additionally, combination therapies, including chemotherapy with nivolumab and ipilimumab, offer promising results. More research is needed to enhance these treatment outcomes.

**Abstract:**

Emerging evidence has shown remarkable advances in the multimodal treatment of esophageal squamous-cell carcinoma. Despite these advances, the oncological outcomes for advanced esophageal cancer remain controversial due to the frequent observation of local recurrence in the regional or other lymph nodes and distant metastasis after curative treatment. For cases of locoregional recurrence in the cervical lymph nodes alone, salvage surgery with lymph node dissection generally provides a good prognosis. However, if recurrence occurs in multiple regions, the oncological efficacy of surgery may be limited. Radiotherapy/chemoradiotherapy can be employed for unresectable or recurrent cases, as well as for selected cases in neo- or adjuvant settings. Dose escalation and toxicity are potential issues with conventional three-dimensional conformal radiotherapy; however, more precise therapeutic efficacy can be obtained using technical modifications with improved targeting and conformality, or with the use of proton beam therapy. The introduction of immune checkpoint inhibitors, including pembrolizumab or nivolumab, in addition to chemotherapy, has been shown to improve the overall survival in unresectable, advanced/recurrent cases. For patients with lymph node recurrence in multiple regions, chemotherapy (5-fluorouracil [5-FU] plus cisplatin) and combination therapy with nivolumab and ipilimumab have shown comparable oncological efficacy. Further prospective studies are needed to improve the treatment outcomes in patients with esophageal cancer with locoregional recurrence.

## 1. Introduction

The number of patients with esophageal cancer has been increasing worldwide in recent years, with the incidence rate of new cases ranked 11th and the mortality rate ranked 7th among all the cancers [1]. Advances in diagnostic technology, particularly in endoscopic methods such as narrowband imaging (NBI) and autofluorescence imaging (AFI), have improved the accuracy of diagnosing esophageal cancer. These advancements have enabled better assessment of T factors in the TNM classification [2]. The treatment options for esophageal cancer include endoscopic resection (ER), radical esophagectomy with lymph node dissection, systemic chemotherapy, and chemoradiotherapy (CRT). For early esophageal cancer confined to the submucosal layer [3], techniques including endoscopic mucosal resection (EMR) and endoscopic submucosal dissection (ESD) have demonstrated efficacy and safety, and they are recommended by the Japan Esophageal Society (JES) [3,4]. For advanced esophageal cancer, treatments vary from surgery or CRT for cancers extending beyond the submucosal layer, but not into the adventitia, to chemotherapy, RT, and CRT for cancers extending beyond the adventitia. If esophageal cancer is resectable but positive for lymph node metastasis, the mainstay treatment is surgery following neoadjuvant chemotherapy (NAC) [4,5]. The preferred surgical approach has shifted from open thoracotomy to minimally invasive esophagectomy (MIE) using thoracoscopy, and robot-assisted MIE (RAMIE) is emerging as a promising technique [6].

Previously, the JES recommended a doublet regimen of cisplatin and 5-fluorouracil (5-FU) as the first-line NAC for advanced esophageal cancer based on the JCOG9907 trial [7]. Recently, this recommendation was updated to a triplet regimen of cisplatin, 5-FU, and docetaxel, reflecting the results of the JCOG 1109 trial [5,8,9]. Despite these advancements, the prognosis for advanced esophageal cancer remains poor, with frequent local recurrence in the regional lymph nodes and distant metastases even after achieving an oncologically curative status. According to the comprehensive registry of esophageal cancer in Japan (2015), the 5-year survival rates of patients with esophageal cancer who undergo esophagectomy vary significantly with the cancer stage, highlighting the ongoing challenges in terms of treatment [10].

Following curative resection, 35–50% of patients develop recurrence, significantly impacting their survival outcomes [11,12,13,14]. Hagens et al. reviewed the distribution pattern of metastatic lymph nodes after surgical resection for esophageal squamous-cell carcinoma (ESCC), revealing specific regions based on the tumor location [15]. The median survival time for patients with recurrent disease, including locoregional lymph node recurrence and distant metastasis, is extremely poor, prompting the need for effective salvage therapies [16,17,18,19,20,21,22]. For clinical stage 0 to III esophageal cancer, therapeutic algorithms have been well established based on endoscopic/surgical resectability and patients’ tolerance of the treatment, with sufficient supportive evidence [4,5]. For unresectable clinical stage IV or recurrent esophageal cancer, recent advances in chemotherapy with or without immune checkpoint inhibitors and the development of definitive CRT represent the available treatment options; however, the oncological results are limited, with weak evidence, and the long-term outcomes are still awaited from the ongoing trials [4,5]. Therefore, the treatment strategy for recurrent esophageal cancer has remained an individual-, tumor-, and patient-tailored approach.

In this review, we focus on the treatment strategies for esophageal cancer with locoregional recurrence after curative treatment, summarizing the recent trends and data. This review aims to address the gaps in the current treatment approaches and provide insights into improving the oncological outcomes for these patients.

## 2. Methods

We conducted a comprehensive literature search on PubMed/MEDLINE using the following terms: ‘esophageal cancer’, ‘lymph node recurrence’, ‘lymphadenectomy’, ‘radiotherapy’, ‘chemotherapy’, and ‘chemoradiotherapy’. Both index terms and free-text words were used. Considering the heterogeneity of the studies, the limited number of enrolled patients, and the specificities of the research question, a narrative review approach was selected. We screened the titles, abstracts, and full-text articles, and we performed a similar search for reference articles. We selected the most relevant, reputable, and up-to-date articles. The publication types included were systematic reviews, meta-analyses, prospective and retrospective studies, case series, and case reports of rare pathologies or novel treatments. Excluded were non-English articles and letters to editors. Due to the expected heterogeneity of the articles, an integrated data analysis was not performed in this narrative review.

## 3. Salvage Lymphadenectomy

Previous retrospective case series have demonstrated that salvage resection has a positive prognostic impact on locoregional lymph node recurrence in ESCC [23,24,25,26]. Particularly, salvage resection yields better long-term outcomes compared to radiotherapy (RT) or CRT, especially when the lymph node recurrence is confined to the cervical lymph nodes [23,25].

However, if the recurrence involves multiple regional lymph nodes, the oncological benefits of upfront surgery may be limited, and salvage surgery can increase the risk of postoperative complications. Shigeno et al. investigated the effectiveness of salvage lymphadenectomy for lymph node recurrence after various treatments, including esophagectomy, ER, CRT, and chemotherapy [27]. They found a complication rate of 30.0% for grade II and 15.0% for grade III or higher (according to the Clavien–Dindo classification [28]), with complications such as postoperative bleeding, lymphatic leakage, and cervical abscess. Other studies have reported postoperative complications, with hoarseness being the most frequent, and overall complication rates ranging from 8.0 to 40%, although the severity was not consistently documented [23,24,25,26].

Regarding operative curability, R0 resection was achieved in 79–85% of patients, irrespective of the site of lymph node recurrence (cervical, mediastinal, or abdominal) or the initial treatment received (esophagectomy, ER, or CRT/chemotherapy). Additionally, 50–79% of patients received neoadjuvant or adjuvant therapy with either CRT, RT, or chemotherapy [23,24,25,27].

In terms of the long-term oncological outcomes, the median survival time (MST) after salvage lymphadenectomy was 15–17 months, consistent with the literature [23,24,25,27], except for one report with five cases showing an MST of 60 months [26]. Shigeno et al. reported five-year overall survival (OS) and recurrence-free survival (RFS) rates of 50.0% and 26.7%, respectively. The survival rates varied slightly based on the initial treatment, with five-year OS and RFS rates of 40.0% and 25.0% after esophagectomy, 75.0% and 33.3% after endoscopic resection, and 50.0% and 25.0% after CRT/chemotherapy, respectively. Additionally, the survival rates were influenced by the stage at initial treatment, with five-year OS and RFS rates of 83.3% and 60.0% for stage I, and 33.3% and 10.0% for stage II or higher, respectively [27]. Nakamura et al. reported a three-year OS rate of 50.7% in patients who underwent salvage lymphadenectomy compared to 26.6% in those who received salvage CRT without surgery [24]. Similarly, Ma et al. found a five-year OS rate of 50.1% in patients who underwent lymphadenectomy compared to 12.6% in those who received salvage RT or CRT, indicating an oncological benefit of salvage surgery over non-surgical treatments [23]. Furthermore, Watanabe et al. reported three-year OS and RFS rates of 75.5% and 51.5%, respectively, with significantly better OS in patients with cervical lymph node recurrence compared to those with mediastinal or abdominal lymph node recurrence [25].

Despite the high rate of R0 resection (79–85%), the re-recurrence rate remained high, ranging from 70 to 74% [24,27]. The re-recurrence sites included the locoregional lymph nodes (21–50%), lymph nodes outside the lymphadenectomy area (15–16%), and distant/organ metastasis (29–37%) [24,27]. The high re-recurrence rate suggests that pathological R0 resection alone does not significantly impact the OS rate [27]. This underscores the need for perioperative CRT/chemotherapy in combination with salvage lymphadenectomy. However, more than half of the patients in the literature received neoadjuvant or adjuvant therapy [23,24,25,27], and the OS rate was not statistically different between patients who underwent salvage lymphadenectomy with or without adjuvant therapy [23].

## 4. Radiotherapy (RT)/Chemoradiotherapy (CRT)

Unlike surgical resection, RT/CRT can preserve the esophageal anatomy and function. Advances in three-dimensional conformal radiation therapy (3DCRT), intensity-modulated radiation therapy (IMRT), and volumetric-modulated arc therapy (VMAT) have enabled more precise treatments by improving the target coverage and conformality. These advancements have enhanced local control of the disease and reduced the exposure of adjacent normal tissues to radiation, thereby minimizing the radiation-induced toxicity [29,30,31]. In Western countries, neoadjuvant CRT followed by surgical resection is commonly recommended as the standard treatment for locally advanced esophageal cancer, with historical trials for neoadjuvant CRT primarily based on 3DCRT [32]. IMRT/VMAT with a high dose and limited volume may be suitable for re-irradiation and can be considered for either primary or recurrent locally advanced esophageal cancer [33].

Previous studies have utilized 3DCRT, IMRT, or VMRT for locoregional recurrence after initial curative treatment. Chen et al. conducted a retrospective study using RT or CRT (mainly VMAT, with a median dose of 60 Gy) in 83 patients with regional lymph node recurrence after radical esophagectomy. The study reported a 12% complete response rate, 69.9% partial response, and three-year and five-year OS rates of 40.1% and 35.1%, respectively, with a median OS of 16 months. There was no significant difference in OS between patients who received CRT and those who received RT alone [34].

Yamashita et al. investigated the oncological outcomes of patients with lymph node oligo-recurrence treated with salvage RT or CRT (median dose of 60 Gy) in a multi-institutional retrospective study [35]. They found three-year OS and RFS rates of 36.7% and 24.1%, respectively, with an MST of 21.6 months. Multivariate analysis indicated that concurrent chemotherapy, a longer disease-free interval, and a smaller maximum lymph node diameter were significantly associated with improved OS. Similarly, Depypere et al. reported the treatment outcomes of patients with clinically isolated locoregional recurrences, finding a five-year OS rate of 23.7% and an MST of 18.7 months. CRT showed superior survival compared to chemotherapy or RT alone when salvage surgery was impossible or contraindicated [36].

Irradiation or re-irradiation for locoregional recurrence must be approached cautiously due to the potential for increased toxicity with dose escalation and a wider radiation field. Chen et al. reported that most toxicities, including esophagitis, neutropenia, and anemia, were grade 1 or 2. Grade 3 toxicities occurred in 18.0% of patients, including skin damage, nausea/vomiting, bronchitis, neutropenia, and radiation pneumonitis [34]. Yamashita et al. reported a 4.6% incidence rate of grade 3 or higher toxicities, including grade 3 anastomotic stenosis, grade 4 cardiac tamponade, hyperglycemia, and esophagobronchial fistula, and grade 5 pneumonia, pleural effusion, mediastinal bronchial fistula, and upper gastrointestinal bleeding [35]. Kim et al. reported severe acute toxicities in patients who underwent re-irradiation for recurrence after primary definitive RT, including three cases of grade 5 esophageal perforation with a tracheoesophageal fistula [37].

Alternatively, proton beam therapy (PBT) may offer disease control for localized recurrent lesions. PBT uses beams of protons or other charged particles to deliver a precise radiation dose to targeted lesions, minimizing the exposure of adjacent normal tissues [38,39,40]. Hiroshima et al. [39] reported the therapeutic efficacy and toxicity of PBT for postoperative lymph node oligo-recurrence in esophageal cancer. They included 11 patients with 13 sites of lymph node oligo-recurrence after curative esophagectomy, excluding those with a history of previous RT. The two-year OS, progression-free survival (PFS), and local control rates were 48.0, 27.3, and 84.6%, respectively, with an MST of 22.4 months. Re-recurrence occurred in 72.7% of patients, primarily outside the irradiated field, except for one case both inside and outside the irradiated field. The observed toxicities were mainly grade 1 or 2 hematological events, with one patient experiencing grade 3 leukopenia. Compared to conventional RT, PBT may be associated with a lower incidence of toxicities and comparable oncological outcomes, suggesting it to be a viable option for treating the locoregional recurrence of esophageal cancer. Most importantly, PBT can reduce the radiation dose to adjacent organs such as the heart and lungs, thereby avoiding serious adverse events, including radiation pneumonitis, pericardial effusion, cardiomyopathy, and arrhythmias [41,42,43,44,45].

## 5. Chemotherapy

The JES recently recommended a triplet regimen of cisplatin, 5-FU, and docetaxel (DCF) as the first-line NAC for locally advanced ESCC. This recommendation follows the JCOG 1109 phase III randomized controlled trial [5,8,9,46], which showed improved survival with DCF compared to cisplatin and 5-FU (CF). The three-year OS and RFS rates were 72.1% and 61.8% for DCF, respectively, compared to 62.6% and 27.7% for CF, respectively [9].

Chemotherapy, with or without radiation, is the standard treatment for unresectable, recurrent, and advanced esophageal cancers [4]. Combination therapy with platinum and fluoropyrimidines is the primary first-line treatment for ESCC, while taxanes and other drugs are used as second-line therapies for patients who relapse or are refractory to first-line treatments [4]. Despite these strategies, the response rate remains poor, with an MST of less than one year [47,48,49,50].

The introduction of immune checkpoint inhibitors, such as pembrolizumab and nivolumab, in addition to chemotherapy, has shown improved outcomes in patients with unresectable, advanced, or recurrent esophageal cancer. ESCC tumors often express programmed death ligand 1 (PD-L1), with its expression detected in approximately 50% of patients with advanced disease [51,52]. Dual checkpoint inhibition with nivolumab and ipilimumab (anti-cytotoxic T-lymphocyte antigen-4 antibodies) has demonstrated longer OS compared to chemotherapy or nivolumab monotherapy in multiple solid tumors [53,54,55]. For patients with lymph node recurrence in multiple regions, chemotherapy (5-FU plus cisplatin) and combination therapy with nivolumab and ipilimumab have shown comparable efficacy. The CheckMate 648 trial [56,57] reported that nivolumab plus chemotherapy (FP: fluorouracil plus cisplatin) significantly improved the OS compared to chemotherapy alone. The median OS was 13.2 months for the nivolumab plus chemotherapy group versus 10.7 months for chemotherapy alone, reducing the risk of death by 26% (hazard ratio, 0.74; *p* = 0.002). Nivolumab plus ipilimumab also led to a longer OS compared to chemotherapy alone, with a median OS of 12.7 months versus 10.7 months, reducing the risk of death by 22% (hazard ratio, 0.78; *p* = 0.01). In patients with tumor-cell PD-L1 expression of 1% or greater, both the OS and PFS were further improved in the nivolumab plus chemotherapy group.

The main results of essential randomized controlled trials using immune checkpoint inhibitors in addition to chemotherapy are as follows:
(1)KEYNOTE-181 [58]: Pembrolizumab significantly prolonged the OS compared to chemotherapy (paclitaxel, docetaxel, or irinotecan) in patients who had received one prior therapy for advanced/metastatic squamous-cell carcinoma or adenocarcinoma of the esophagus.(2)KEYNOTE-590 [59]: Pembrolizumab plus chemotherapy (FP) significantly prolonged the OS and PFS in patients with untreated, advanced, unresectable, or metastatic esophageal cancer or Siewert type 1 gastro-esophageal cancer.(3)ATTRACTION-3 [52]: Nivolumab significantly improved the OS compared to chemotherapy (paclitaxel or docetaxel) in patients with unresectable, advanced, or recurrent ESCC.(4)ESCORT-1st [60]: Camrelizumab plus chemotherapy (TP: paclitaxel plus cisplatin) resulted in longer OS and PFS when used as the first-line treatment in patients with untreated, advanced, or metastatic ESCC.(5)JUPITER-06 [61]: Toripalimab plus chemotherapy (TP) significantly improved the OS and PFS in patients with untreated advanced ESCC.(6)ORIENT-15 [62]: Sintilimab plus chemotherapy (TP or FP) significantly prolonged the OS and RFS in patients with ESCC who had not received prior systemic therapy.

A summary of these trials is shown in Table 1.

## 6. Limitations

This study has several limitations. First, our review exclusively focuses on ESCC. Different histological types, such as adenocarcinoma, may exhibit different malignant behaviors, including varying frequencies of lymph node metastasis, preferred metastatic sites, and sensitivity to chemotherapy, radiotherapy, or immunotherapy. Second, ESCC is predominant in East Asia and Africa, while esophageal adenocarcinoma accounts for the majority of cases in Europe and North America. Consequently, different classifications of esophageal cancer are used in Asia (Japan) [3,46] and America [2], which may limit the applicability of the clinical trial results to different populations. Third, the rarity of recurrent esophageal cancer and the heterogeneity of prior treatment modalities result in a lower level of available evidence for research on its treatment. Fourth, this narrative review provides qualitative summaries of the treatment but does not provide quantitative comparisons using a meta-analytic approach due to the heterogeneities in the studies and the limited data availability.

## 7. Future Directions and Conclusions

While some cases of locoregional lymph node recurrence of ESCC can be cured with a single type of treatment, most patients with high recurrence rates require multimodal therapy. In Japan, there are case reports of complete remission after multidisciplinary treatment for lymph node recurrence or metastasis [63]. Recently, a randomized controlled trial reported improved overall survival with neoadjuvant CRT and adjuvant therapy using immune checkpoint inhibitors (nivolumab) following esophagectomy [64]. This underscores the potential benefit of incorporating neo- or adjuvant immune checkpoint inhibitors with chemotherapy/CRT to enhance the oncological outcomes of advanced ESCC. Future studies are expected to demonstrate the potential benefit of immune checkpoint inhibitors for patients with low or negative expression of PD-L1. Investigations of the tumor response, safety/adverse events, and long-term survival are warranted based on the ongoing trials. Although previous studies have demonstrated improved perioperative outcomes and the equivalent oncological results of MIE or RAMIE compared with open esophagectomy for the initial treatment of esophageal cancer, the role of minimally invasive approaches for recurrent ESCC has scarcely been discussed. Furthermore, it should be noted that the treatment strategies for locoregionally recurrent ESCC are identical to that of oligometastases of ESCC [65]. Finally, we propose an algorithm of the treatment strategies for locoregional recurrence of ESCC (Figure 1). Further prospective studies are needed to improve the treatment outcomes in patients with esophageal cancer experiencing locoregional recurrence.

## Figures and Tables

**Figure 1 cancers-16-02539-f001:**
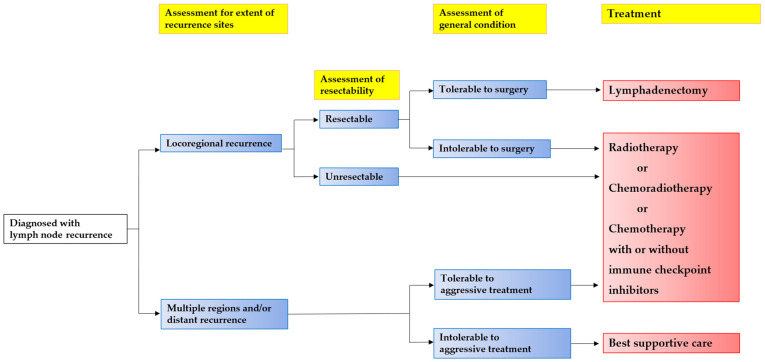
Suggested algorithm of the treatment strategies for lymph node recurrence in esophageal squamous-cell carcinoma.

**Table 1 cancers-16-02539-t001:** Summary of clinical trials using chemotherapy with immune checkpoint inhibitors.

Trial Name	Enrollment Criteria	Immune Checkpoint Inhibitor (Line of Treatment)	Treatment Arm	Median OS (Months)	HR	Median PFS (Months)	HR
CheckMate 648 [56,57]	Unresectable, recurrent, or metastatic disease: SCC or adenosquamous-cell carcinoma	Nivolumab (first line)	Nivolumab + ipilimumab (*n* = 131) vs. chemotherapy alone (*n* = 137)	12.7 vs. 10.7 *p* = 0.01	0.78	2.9 vs. 5.6 *p*-value not tested	1.26
			Nivolumab + chemotherapy (*n* = 126) vs. chemotherapy alone (*n* = 137)	13.2 vs. 10.7 *p* = 0.002	0.74	5.8 vs. 5.6 *p* = 0.04	0.81
KEYNOTE-181 [58]	Unresectable, recurrent, or metastatic disease: SCC or adenocarcinoma	Pembrolizumab (second line)	Pembrolizumab (*n* = 314) vs. chemotherapy (PD-L1 combined positive score ≥ 10) (*n* = 314)	9.3 vs. 6.7 *p* = 0.074	0.69	2.1 vs. 3.4 *p*-value not tested	1.11
KEYNOTE-590 [59]	Unresectable or metastatic disease: SCC or adenocarcinoma	Pembrolizumab (first line)	Pembrolizumab + chemotherapy (*n* = 373) vs. chemotherapy alone (*n* = 376)	12.4 vs. 9.8 *p* < 0.0001	0.73	6.3 vs. 5.8 *p* < 0.0001	0.65
ATTRACTION-3 [52]	Unresectable, recurrent, or metastatic disease: SCC or adenosquamous-cell carcinoma	Nivolumab (second line)	Nivolumab (*n* = 210) vs. chemotherapy (*n* = 209)	10.9 vs. 8.4 *p* = 0.019	0.77	1.7 vs. 3.4 *p*-value not tested	1.08
ESCORT-1st [60]	Unresectable, recurrent, or metastatic disease: SCC	Camrelizumab (first line)	Camrelizumab + chemotherapy (*n* = 298) vs. placebo + chemotherapy (*n* = 298)	15.3 vs. 12.0 *p* = 0.001	0.70	6.9 vs. 5.6 *p* < 0.001	0.56
JUPITER-06 [61]	Unresectable, recurrent, or metastatic disease: SCC	Toripalimab (first line)	Toripalimab + chemotherapy (*n* = 257) vs. placebo + chemotherapy (*n* = 257)	17.0 vs. 11.0 *p* = 0.0004	0.58	5.7 vs. 5.5 *p* < 0.0001	0.58
ORIENT-15 [62]	Unresectable, recurrent, or metastatic disease: SCC	Sintilimab (first line)	Sintilimab + chemotherapy (*n* = 327) vs. placebo + chemotherapy (*n* = 332)	16.7 vs. 12.5 *p* < 0.001	0.63	7.2 vs. 5.7 *p* < 0.001	0.56

SCC: squamous-cell carcinoma, OS: overall survival, HR: hazard ratio, PFS: progression-free survival.

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
