# Peer review of "Treatment Strategies for Locoregional Recurrence in Esophageal Squamous-Cell Carcinoma: An Updated Review"

_cancers, 2024, doi:10.3390/cancers16142539_

Round 1
Reviewer 1 Report
Comments and Suggestions for Authors
The manuscript discusses various treatment strategies for locoregional ESCC comprehensively. The review will be more interesting if the discussion on the efficacy and limitations of each treatment in recurred ESCC is included in the manuscript.
In a few places, many words are together (punctuation problem), please check and correct it.
Please also comment on the therapy for oligometastatic tumors because the treatment is the same for both.
Author Response
- The manuscript discusses various treatment strategies for locoregional ESCC comprehensively. The review will be more interesting if the discussion on the efficacy and limitations of each treatment in recurred ESCC is included in the manuscript.
Response:
Thank you for thoughtful comments. We have added several sentences to ‘Future direction and conclusions’ section regarding unsolved issues and future expectations in each treatment options for recurrent ESCC (Line 289-296).
- In a few places, many words are together (punctuation problem), please check and correct it.
Response:
Thank you for your advice. For more refined grammatical correction, we have utilized the MDPI English editing service before resubmitting the revised manuscript (english-82518).
- Please also comment on the therapy for oligometastatic tumors because the treatment is the same for both.
Response:
We agree with that this point should be mentioned. We have cited 1 paper (Ref. No. 65) and added the description in “Future directions and conclusion sections” (Line 296-298).
Reviewer 2 Report
Comments and Suggestions for Authors
This is a comprehensive review of ESCC treatment strategies with an informative potential for clinicians. Here are a couple of suggestions:
1. In the section on methods, please provide the numbers of published articles included and excluded, as well as the reasons for exclusion and inclusion.
2. The manuscript is hard to read in its current format, please ask someone with a better English background to look it over before resubmission.
Comments on the Quality of English LanguageThe manuscript is hard to read in its current format, please ask someone with a better English background to look it over before resubmission.
Author Response
This is a comprehensive review of ESCC treatment strategies with an informative potential for clinicians. Here are a couple of suggestions:
- In the section on methods, please provide the numbers of published articles included and excluded, as well as the reasons for exclusion and inclusion.
Response:
We are grateful to the reviewer for carefully reading the manuscript. As we mentioned in the introduction, the prognosis of advanced esophageal squamous cell carcinoma (ESCC) remains poor because 35-50% of patients develop locoregional recurrence and/or distant metastasis after initially curative treatment. The prognosis of recurrent esophageal cancer is even worse and established treatment guidelines with higher evidence level are still lacking.
Under these circumstances, we did our best to confront with locoregionally recurrent ESCC by collecting most relevant, reputable, and up-to-date articles including practice guidelines, systematic reviews, meta-analyses, prospective and retrospective studies, case series, and case reports of rare pathologies or novel treatments. Considering the heterogeneity of the studies, limited number of enrolled patients, and the specificities of the research question, a narrative approach was selected for this topic. Therefore, it is very difficult to provide strict criteria of inclusion and exclusion as well as the number of screened articles in accordance with systematic approach (PRISMA diaphragm).
We have reframed some descriptions in the Methods and Limitations sections accordingly (Line92-3, 95-96, 98-100, 276-279). Additional sentences were highlighted in red and underlined. The revised manuscript was re-edited by MDPI Editing service (english-82518) before submission.
- The manuscript is hard to read in its current format, please ask someone with a better English background to look it over before resubmission.
Response:
Thank you for your advice. We have utilized the MDPI English editing service before resubmitting the revised manuscript (english-82518).
Reviewer 3 Report
Comments and Suggestions for Authors
Dear Editor,
I was pleased to review the manuscript entitled ''Treatment Strategies for Locoregional Recurrence in Esophageal Squamous Cell Carcinoma: An Updated Review''. Mitamura et al. examined locoregional recurrence, which is an important problem in the treatment of squamous cell esophageal cancer. The review was well summarized according to lymph node involvement and further prospective studies are needed to improve treatment outcomes in esophageal cancer patients with locoregional recurrence. Unfortunately, I think the manuscript is not of sufficient quality to be published.
Sincerely
Comments on the Quality of English LanguageMinor editing of English language required
Author Response
Dear Editor,
I was pleased to review the manuscript entitled ''Treatment Strategies for Locoregional Recurrence in Esophageal Squamous Cell Carcinoma: An Updated Review''. Mitamura et al. examined locoregional recurrence, which is an important problem in the treatment of squamous cell esophageal cancer. The review was well summarized according to lymph node involvement and further prospective studies are needed to improve treatment outcomes in esophageal cancer patients with locoregional recurrence. Unfortunately, I think the manuscript is not of sufficient quality to be published.
Sincerely
Response:
Thank you for your recognition and kind comments. We have utilized the MDPI English editing service before resubmitting the revised manuscript (english-82518).
Reviewer 4 Report
Comments and Suggestions for Authors
Manuscript entitled "Treatment Strategies for Locoregional Recurrence in Esophageal Squamous Cell Carcinoma: An Updated Review"
Major issues:
1. The authors should summarize the enrollment criteria of each clinical trial in Table 1.
2. The authors should mention more about the current status of ESCC treatment regarding the approaches and limitations.
3. The authors should compare the survival benefits of each trial using a meta-analysis strategy.
Comments on the Quality of English LanguageAcceptable
Author Response
Manuscript entitled "Treatment Strategies for Locoregional Recurrence in Esophageal Squamous Cell Carcinoma: An Updated Review"
Major issues:
- The authors should summarize the enrollment criteria of each clinical trial in Table 1.
Response:
Thank you for your valuable suggestion. We have added “enrollment criteria” to Table 1. To further highlight the study protocol of each trial, we have also added “number of patients” and “line of treatment”.
- The authors should mention more about the current status of ESCC treatment regarding the approaches and limitations.
Response:
Thank you for carefully reading the manuscript. Unlike resectable esophageal cancers, treatment options are still limited without the strong evidence of long-term results despite the recent advances in surgical approach, chemotherapy and radiotherapy. We have rephrased several sentences in the introduction and future directions/conclusions. Additional sentences were highlighted in red and underlined (Line 75-83, 289-296).
- The authors should compare the survival benefits of each trial using a meta-analysis strategy.
Response:
Thank you for this important comment. However, this narrative review provides qualitative summaries of the treatment, but does not provide quantitative comparisons using integrated statistical analysis (meta-analytic approach) due to the heterogeneity in the studies and limited data availability.
We have reframed some descriptions in the Methods and Limitations sections accordingly (Line92-3, 95-96, 98-100, 276-279).
Round 2
Reviewer 2 Report
Comments and Suggestions for Authors
It is acceptable to me.
Reviewer 3 Report
Comments and Suggestions for Authors
Dear Editor,
Unfortunately, the presented study does not provide new information and approaches to the literature.
Sincerely
Comments on the Quality of English LanguageMinor editing of English language required
Reviewer 4 Report
Comments and Suggestions for Authors
The revision is acceptable.
Comments on the Quality of English LanguageAcceptable